# Towards the Clinical Application of Gene Therapy for Genetic Inner Ear Diseases

**DOI:** 10.3390/jcm12031046

**Published:** 2023-01-29

**Authors:** Ghizlene Lahlou, Charlotte Calvet, Marie Giorgi, Marie-José Lecomte, Saaid Safieddine

**Affiliations:** 1Institut Pasteur/Institut de l’Audition, Technologie et Thérapie Génique de la Surdité, Sorbonne Université, INSERM, 75012 Paris, France; 2Département d’Oto-Rhino-Laryngologie, Unité Fonctionnelle Implants Auditifs, Groupe Hospitalo-Universitaire Pitié-Salpêtrière, APHP Sorbonne Université, 75013 Paris, France; 3Zurich Integrative Rodent Physiology (ZIRP), University of Zurich, CH-8057 Zurich, Switzerland; 4Centre National de la Recherche Scientifique, 75016 Paris, France

**Keywords:** gene therapy, hearing impairment, vestibular defect, AAV

## Abstract

Hearing loss, the most common human sensory defect worldwide, is a major public health problem. About 70% of congenital forms and 25% of adult-onset forms of deafness are of genetic origin. In total, 136 deafness genes have already been identified and there are thought to be several hundred more awaiting identification. However, there is currently no cure for sensorineural deafness. In recent years, translational research studies have shown gene therapy to be effective against inherited inner ear diseases, and the application of this technology to humans is now within reach. We provide here a comprehensive and practical overview of current advances in gene therapy for inherited deafness, with and without an associated vestibular defect. We focus on the different gene therapy approaches, considering their prospects, including the viral vector used, and the delivery route. We also discuss the clinical application of the various strategies, their strengths, weaknesses, and the challenges to be overcome.

## 1. Introduction

Deafness, with and without associated balance defects, is the most common sensory disorder in humans. It is a major public health issue, affecting people of all ages. According to the World Health Organization, 466 million people worldwide, including 34 million children, have disabling hearing impairment. The cause of hearing loss is genetic in 70% of congenital cases and 25% of adult-onset cases [1,2]. Congenital hearing loss has a negative impact on children’s lives, causing delays in speech and language acquisition, and social and emotional development problems [1,3]. In adults, hearing impairment is associated with social and psychological difficulties and with more severe cognitive decline in the elderly [4,5]. Balance disorders due to vestibular dysfunctions result in gaze instability, which worsens during movement, significantly decreasing participation in physical activities, and having major consequences for the personal and professional lives of those affected [6].

There is currently no specific cure for inner ear disorders. Patients with sensorineural hearing loss benefit from hearing rehabilitation with conventional hearing aids for milder cases, and cochlear implants for the most severe cases. Vestibular disorders are mostly managed by vestibular physiotherapy, which provides central compensation. 

Improvements in our understanding of the genetics of hearing impairment over the last four decades have generated a growing interest in gene therapy for inner ear defects among researchers, clinicians, public and private funding organizations, industry, and patients. This keen interest is due to several preclinical studies on animal models of human genetic hearing loss showing promising results for a potential treatment of genetic causes of deafness [7,8,9,10,11,12,13]. These successes have laid the foundations for the future use of gene therapy to treat hearing loss and vestibular disorders of genetic origin. However, many challenges remain in terms of treatment efficacy, administration techniques, and safety. 

We discuss here gene therapy approaches for alleviating and/or curing monogenic inherited hearing and balance impairments. We report recent trends in inner gene therapy, including the latest approaches being developed and improvements in therapeutic strategy that may be approved in the near future. We also discuss the additional challenges that must be overcome to ensure the safe and effective transfer of these therapies to humans.

## 2. Gene Therapy Strategies

The perspectives for gene therapy have broadened with respect to its initial definition: the treatment of a genetic disease by the replacement of a defective gene with a functional copy by gene transfer. Gene therapy is now considered to be the transfer of a nucleic acid, either DNA or RNA, to treat or prevent a disorder through various strategies, including gene replacement (or gene augmentation), gene silencing, and base and prime editing/in situ repair [14,15]. The challenges inherent to all these approaches are similar and include the specific, safe, and efficient delivery of the therapeutic material (DNA, RNA, oligonucleotides, siRNA, or the molecular elements of the CRISPR/Cas9 system) into target cells, mostly with the aid of a modified viral vector (as discussed below), or, more rarely, non-viral vectors, such as liposomes. 

Various efforts have been made to treat or prevent hearing loss in mouse models of human deafness forms, with various vectors used to transfer the genetic material into the inner ear (see Table 1). We will focus here on the most recent and promising investigations, some of which have now progressed beyond the preclinical stage, and renewed interest in gene therapy for hearing loss, both in industry and in academic laboratories engaged in translational research.

### 2.1. Gene Replacement Prevents and Cures Congenital Deafness

The gene replacement strategy involves delivering a corrected copy of the gene responsible for deafness, to supplement the function of a non-functional mutant gene [36]. This is the strategy of choice for treating inherited disorders of the inner ear in which protein function is lost: autosomal recessive diseases (DFNB deafness and syndromic recessive deafness) and autosomal dominant diseases displaying haploinsufficiency (i.e., the product of the single functional allele is insufficient to ensure normal cell function). Various degrees of success have been achieved in preclinical investigations of gene therapy based on gene replacement strategies in mouse models of human deafness. These studies are summarized in Table 1. Outcomes are variable mostly due to the therapeutic time window and/or low transduction rates of target cells. The efficacy of gene therapy for hereditary deafness and balance disorders has been evaluated in murine models at different stages of inner ear maturation, from in utero to after the onset of hearing [12,16], but most studies to date were performed at early neonatal stages [7,10,20,22,26,27,31,32,33]. These studies have raised hopes for clinical trials in patients with hearing and balance disorders in the near future, despite the modest nature of some results obtained.

### 2.2. RNA Interference Therapy for Hearing Loss

RNA interference (RNAi) is a natural posttranscriptional process for the regulation of gene expression in eukaryote cells [37]. The discovery of RNAi has considerably extended the field of gene therapy to include RNA as targets [38]. RNA-based therapy involves the use of microRNAs, siRNAs, or antisense oligonucleotides (ASOs) designed on the basis of sequence complementarity to interfere with the posttranslational regulation process and silence harmful mRNAs. RNAi can be triggered by synthetic nucleotides and can, therefore, interfere with almost any gene of interest, including those difficult to target selectively with pharmaceutical approaches [39,40].

Almost 80% of autosomal-dominant forms of non-syndromic hearing loss in humans are caused by disruptive point mutations resulting in potential gain-of-function or dominant negative mutations, for which mRNA-based gene-silencing technology is highly suitable. This approach can be used to silence the dominant allele without altering the expression of the wild-type gene.

One proof-of-principle study by Maeda et al. [41] focused on DFNA3 hearing loss, which is caused by dominant mutations of the *GJB2* gene [42,43] encoding the transmembrane protein connexin 26 (CX26) expressed by the cochlear supporting cells [44]. Maeda et al. showed that the use of siRNA in vivo in mice selectively suppressed expression of the mutant *Gjb2*_R75W_ allele, which causes hearing loss through a dominant-negative effect, without significantly affecting expression of the wild-type *Gjb2* allele [41]. This treatment prevented the hearing loss otherwise associated with the R75W dominant-negative variant of CX26.

An elegant RNAi approach based on an artificial microRNA (miRNA) was recently shown to rescue the progressive hearing-loss phenotype of the Beethoven (*Bth*) mouse, a model of the DFNA36 human autosomal-dominant non-syndromic form of deafness. This mouse carries a dominant gain-of-function mutation of *Tmc1* (transmembrane channel-like 1) [22]. A specific siRNA sequence for silencing this gene was designed and embedded in an artificial miRNA scaffold for delivery to the cochlea of neonatal or adult *Tmc1^Bth^*^/+^ with an associated adenovirus (AAV) vector. This selectively suppressed the *Tmc1* c.1235T > A (p.Met412Lys) dominant gain-of-function allele and prevented the development of profound hearing loss [22] (Table 1). These observations highlight the potential of allele-specific RNAi-based therapeutic approaches to mitigate sensorineural deafness caused by dominant-negative mutations, particularly as this type of auditory deficit is postlingual and progressive, providing a large therapeutic time window for intervention.

### 2.3. Gene-Editing Therapy for Hearing Loss

The advent of gene-editing technologies based on the clustered regularly interspaced short palindromic repeat (CRISPR)-Cas9 endonucleases is revolutionizing gene therapy by making it possible to perform manipulations highly efficiently at any endogenous locus, thereby facilitating gene manipulation in situ. This technology largely outperforms ex vivo homologous recombination and techniques based on TALENS and zinc finger nucleases [45,46]. The CRISPR/Cas9 system was first discovered in bacteria, in which it serves as an adaptive immune system against invasive viral genomes [47]. One of the principal reasons for the success and efficiency of CRISPR/Cas9 is its high specificity for the target sequence, similar to that of the RNAi approach [47,48,49]. Like RNAi approaches, it also uses synthetic RNAs specific to the target sequence (short-guide RNAs), which are delivered together with a Cas9 enzyme to mediate the creation of specific double-strand breaks in the target DNA sequence. The DNA repair pathway operates by frameshift and/or stop codon mutations [50]. Gene editing with CRISPR/Cas9 provides an interesting approach complementary to RNAi for the treatment of autosomal-dominant hearing loss due to disruptive point mutations. Gene editing was recently shown to prevent hearing loss in the Beethoven mouse model of DFNA36 [9,23] (Table 1), as previously reported for RNAi. A single injection of Cas9 plus the guide RNA mixture selectively suppressed the dominant gain-of-function mutation in Beethoven mice and prevented progressive deafness in the newborn mutant mice. Results for the correction of point mutations by CRISPR/Cas gene editing are promising, but the efficiency of homologous direct repair remains low and restricted to mitotic cells [51], and is generally surpassed by non-homologous end joining junction (NHEJ), which frequently leads to unwanted indel outcomes [52,53]. However, NHEJ occasionally leads to indels that coincide with the desired editing outcomes. This was the case for an in vitro frameshift repair using a strategy based on the gRNA-Cas9-induced precise cleavage and the NHEJ-mediated highly biased editing without a template [54,55,56]. Remarkably, using this strategy in vivo, Liu et al. were able to restore hearing and balance function in a mouse model of human deafness DFNB23, which is due to a spontaneous single nucleotide insertion causing a frameshift in the 7.9 kb Pcdh15 transcript [35]. The NHEJ-mediated frame restoration strategy is attractive for developing treatments for frameshift mutations. However, the frequency of indel byproducts remains high when considering a clinical application scenario. Further improvements are needed, including Cas9 and RNA guide optimization, delivery to the inner ear target cells, and precisely controlled indels. Several studies have shown that point mutations can be corrected, but they were often associated with aberrant editing within the target site itself, i.e., in which CRISPR-Cas is designed to initiate the repair process [57,58,59].

To overcome the major challenges and limitations of the CRISPR/Cas use for the treatment of human diseases, extensive CRISPR/Cas protein complex engineering led to the development of “base editing” and “prime editing” strategies [15]. The base editing approach allows the precise conversion of a single base into another in a genomic DNA with relatively high efficiencies and without generating double-strand breaks or exogenous donor DNA templates, minimizing the unwanted indels [60,61,62]. Remarkably, optimized cytosine base editors virally delivered in vivo to the inner ear corrected the Tmc1 C.A545G, a deafness point mutation, to wild-type sequence (c.A545A), leading to restored sensory transduction and improved auditory function in the treated mouse [21]. This discovery provides strong evidence in favor of the development of a base editing strategy to correct virtually all kinds of deafness mutation.

The prime editing strategy consists in directing the prime editing molecular machinery to the genomic region to be corrected and replacing it with the desired sequence. The prime editing molecular machinery involves a unique guide RNA, known as the prime editing gRNA (pegRNA), which guides a catalytically impaired Cas9 endonuclease fused to a reverse transcriptase enzyme (RT) [15]. Once the endonuclease notches the polynucleotide strand to be replaced, the RT generates in situ the correct complementary DNA using the pegRNA as a template. The proof of concept for the in vivo effectiveness of prime editing has recently been provided in a mouse model of type I tyrosinemia diseases [63,64]. To date, there are no in vivo data available concerning the field of genetic deafness; however, the prime editing is in remarkable and constant optimization, including precision, efficiency, specificity, and safety, which leaves the option of an eventual clinical application being conceivable for genetic deafness.

## 3. Viral Vectors

### 3.1. Adenovirus and Lentivirus

Despite the safety issues associated with adenoviruses (AdVs), such as toxicity and immune responses, these vectors are being tested in a large number (~20%) of clinical trials currently underway around the world [65,66]. The first phase I clinical trial for the treatment of deafness by gene therapy methods used AdVs (NCT02132130). However, a functional defect of the outer hair cells (OHC) was reported upon AdV delivery to the cochlea [67,68,69], making the use of these viruses as vectors for gene transfer to the inner ear challenging. However, recent advances in adenovirus engineering have significantly decreased their immunogenicity, extended transgene expression, and improved safety [70], which remains to be demonstrated for the inner ear.

Lentiviruses (LVs), which belong to a subclass of Retroviridae, have emerged as another possible vehicle for gene delivery applications. One of the principal obstacles to their use for in vivo gene therapy is the risk of insertional mutagenesis, which could disrupt gene function in the transduced cells [71]. Although several safer integrase-defective lentiviral vectors (IDLVs) have since been engineered [72,73], and the third generation of these vectors is now widely used [71]. Interestingly, LV vectors have been shown to deliver *Myo7A* cDNA efficiently and to mediate the correction of several abnormal retinal phenotypes in mouse models of Usher syndrome type 1B. These results represent a major step forward in the development of LV-mediated gene therapy for the treatment of Usher 1B blindness [74,75]. It remains unclear whether lentivirus-mediated gene therapy would be able to rescue the inner ear defect of Usher syndrome type 1B. Proofs-of-concept for the consistent transduction of inner ear tissues with LVs are currently lacking [76]. Attempts are currently being made to improve the specificity of LVs and their targeting of various cell types [77]. LV variants efficiently targeting the inner ear sensory cell will probably follow, together with the successful establishment of gene therapy to treat deafness with LV technology.

### 3.2. Adeno-Associated Virus

AAV, a single-stranded DNA virus, is one of the smallest known viruses and is among the potential gene therapy vehicles most actively investigated for therapeutic strategies [78,79]. AAV-based gene therapies have been used clinically in a number of situations, with remarkable therapeutic benefits and an excellent safety record [79]. These vectors are frequently chosen for gene therapy because they are highly stable, non-pathogenic, and capable of infecting diverse cells without undergoing site-specific integration into the chromosomes of the target cell [80,81]. To date, 13 naturally occurring AAV serotypes and over 108 AAV variants have been identified or engineered, and this number will undoubtedly continue to grow, as will the number of cell types that can be targeted [82]. Unsurprisingly, AAVs have emerged as the vector of choice for inner ear gene delivery in vivo. Indeed, it is becoming increasingly clear that the considerable diversity of the cells in the inner ear potentially affected during hearing loss will require the use of more than one vector.

Transduction rates and the cell types transduced in the inner ear can be influenced by several factors, including the AAV capsid/promoter combination, virus preparation and purification methods, the route and technique of administration, and the dose administered [81,83,84,85,86]. Furthermore, AAV cell tropism and transduction rate can vary between cell types, animal species, and inner ear development stages [84]. Many studies have investigated the tropism of rAAV in inner ear cells (mainly cochlear cells) in vivo in mice, and, more recently, in non-human primates (NHP). The results obtained in mice are summarized in Table 2.

A few recent studies have focused on the transduction rates and profiles obtained in NHP. Two modified AAV9 vectors (AAV9.PHP.B and AAV-S) robustly transduced both hair cells and supporting cells towards the cochlea after injection through the round window membrane (RWM) [33,87,88]. Following injection of the Anc80L65 vector through the RWM with oval window fenestration, Andres-Mateos et al. reported transduction rates of up to 90% for inner hair cells (IHCs), with a more variable percentage of the cells transduced in vestibular organs, in which transduction rates were much lower with AAV1 (up to 30% of IHCs in the apical region) [89]. Interestingly, a recent study found a transduction of IHC and spiral ganglion cells after cerebro-spinal fluid delivery of numerous AAV serotypes variants (AAV1, AAV2, and AAV9) to a *Chorocebus aethiops* (African green) monkey [90]. The common finding that emerges from these cell tropism studies is that most mature OHCs are refractory to transduction by existing AAV serotypes. It will therefore be essential to identify or design serotypes capable of targeting these cells to ensure the complete restoration of hearing.

**Table 2 jcm-12-01046-t002:** Results of viral transduction after the injection into the inner ear of several AAV serotypes driving the expression of eGFP under the control of a constitutive promoter.

Capsid	Injection Stage	Injection Route	Inner Ear Hair Cell Transduction (%)			Transduction of Other Inner Ear Cells	References
			IHC	OHC	VHC		
AAV1	Neonatal	RW	0–67	0–14	0	Inner phalangeal cells and Deiters’ cells	Askew et al., 2015 [19]; György et al., 2017 [24]; Landegger et al., 2017 [91]; Pan et al., 2017 [27]; Emptoz et al., 2017 [7]
		CO	36	17	NR	Marginal cells	Chang et al., 2015 [26]; György et al., 2017 [24]
	Mature	RW with PSCC fenestration	10	<5	<10	Stria vascularis cells	Omichi et al., 2020 [92]
		PSCC	6	0	NR	NR	Tao et al., 2018 [93]
AAV2	Neonatal	RW	0–78	0–50	NR	NR	Emptoz et al., 2017 [7]; Askew et al., 2015 [19]; Landegger et al., 2017 [91]; Geng et al., 2017 [94]
		PSCC	44	54	NR	Pillar cells	Isgrig et al., 2019 [95]
	Mature	RW with PSCC fenestration	95	80	0	0	Omichi et al., 2020 [92]
		PSCC	85	10	7	NR	Tao et al., 2018 [93]
AAV5	Neonatal	RW	0	0	0	Supporting and mesothelial cells	Emptoz et al., 2017 [7]
		CO	0	0	0	Supporting, mesothelial, and Reissner’s membrane cells	Iizuka et al., 2015 [96]
AAV6	Neonatal	RW	15–20	5–10	NR	NR	Askew et al., 2015 [19]; Landegger et al., 2017 [91]
	Mature	PSCC	5	0	NR	NR	Tao et al., 2018 [93]
AAV8	Neonatal	RW	10–90	5–28	90	Spiral ganglion neurons	Askew et al., 2015 [19]; Chien et al., 2015 [97]; Emptoz et al., 2017 [7]; Landegger et al., 2017 [91]; Geng et al., 2017 [94]; Dulon et al., 2018 [32]; Xia et al., 2012 [98]
		PSCC	49–86	13–52	53	Marginal, vestibular supporting, and pillar cells	Isgrig et al., 2019 [95]; Guo et al., 2017 [99]
	Mature	RW with PSCC fenestration	90	<10	35	Stria vascularis cells and spiral ganglion neurons	Omichi et al., 2020 [92]
		PSCC	75	0	41	NR	Tao et al., 2018 [93]
AAV9	Neonatal	RW	5	5	NR	NR	Askew et al., 2015 [19]
		CO	57	15	12	NR	Gu et al., 2019 [100]
	Mature	RW + PSCC	95	<5	good	NR	Yoshimura et al., 2018 [101]
		RW	30	0	0	NR	Yoshimura et al., 2018 [101]
		RW with PSCC fenestration	100	0	20	Stria vascularis cells, and spiral ganglion neurons	Omichi et al., 2020 [92]
		PSCC	60	0	20	NR	Tao et al., 2018 [93]
Anc80L65	Neonatal	RW	90–100	80–95	95	Pillar and Deiters’ cells	Pan et al., 2017 [27]; Landegger et al., 2017 [91]; Lee et al., 2020 [86]
		CO	100	90	NR	Supporting cells	Gu et al., 2019 [100]
		PSCC	94	67	NR	Pillar cells	Isgrig et al., 2019 [95]
		Utricle	100	30–90	robust	Pillar and Deiters’ cells	Lee et al., 2020 [86]
	Mature	PSCC post	95–100	40–50	40	NR	Suzuki et al., 2017 [102]; Tao et al., 2018 [93]
		RW + PSCC	90	-	good	NR	Yoshimura et al., 2018 [101]
		RW with PSCC fenestration	100	50	35	Stria vascularis cells, and spiral ganglion neurons	Omichi et al., 2020 [92]
		Utricle	100	0–20	moderate	NR	Lee et al., 2020 [86]
AAV2 quadY-F	Mature	RW	85	NR	NR	NR	Akil et al., 2019 [13]
AAV2.7m8	Neonatal	PSCC	84	83	NR	Pillar and internal phalangeal cells	Isgrig et al., 2019 [95]
		Utricle	40–100	40	NR	NR	Lee et al., 2020 [86]
	Mature	RW	84	75	NR	NR	Isgrig et al., 2019 [95]
AAV8BP2	Neonatal	PSCC	56	44	NR	NR	Isgrig et al., 2019 [95]
AAV9-PHP.B	Neonatal	RW	70–100	35–70	NR	NR	György et al., 2019 [33]; Lee et al., 2020 [86]
		Utricle	100	100	robust	NR	Lee et al., 2020 [86]
	Mature	PSCC	100	0	robust	NR	György et al., 2019 [33]
		Utricle	100	20–80	robust	NR	Lee et al., 2020 [86]
AAVrh.39	Mature	PSCC	55	0	NR	NR	Tao et al., 2018 [93]
AAVrh.43	Mature	PSCC	95	0	NR	NR	Tao et al., 2018 [93]
AAV-S	Neonatal	RW	100	50–75	robust	Interdental, inner and outer sulcus, Claudius cells, and spiral ganglion neurons	Ivanchenko et al., 2021 [88]
AAV-ie	Neonatal	RW	100	60–100	100%	All cell types of supporting cells	Tan et al., 2019 [103]

CO = cochleostomy; IHC = inner hair cells; NR = not reported; OHC = outer hair cells; PSCC = posterior semicircular canal; RW = round window; VHCs = vestibular hair cells.

One of the drawbacks of AAVs, limiting their utility for gene therapy for deafness, is their limited packaging capacity of 4.7 kb, below the size of most of the identified deafness genes. Several research groups have attempted to overcome this limitation by adopting an approach to double AAV packaging capacity by splitting a large transgene into two fragments, each packaged in a different AAV vector. The full-length expression cassette is reconstituted upon co-infection of the same cell with the two AAV vectors containing the fragments [104,105,106,107]. Such dual AAV gene therapy was recently adapted for use with the otoferlin gene, defects of which underlie DFNB9, one of the most frequent prelingual forms of non-syndromic human deafness [108,109,110,111]. The otoferlin cDNA, split between two AAVs delivered together to the cochlea of DFNB9 mice, restored hearing in these otoferlin-null mice [13].

## 4. Routes for Inner Ear Gene Delivery

One of the major challenges for gene delivery to the inner ear is the anatomic, physiological, and cellular barriers present in this structure. The inner ear is a small but complex organ, with a highly regulated environment, encased in the temporal bone. The membranous labyrinth, filled with endolymph and located within the otic capsule, holds the cochlear and vestibular sensory organs: the two main targets of gene therapy for inner ear disorders. The space between the membranous labyrinth and the otic capsule is filled with perilymphatic fluid, which has a composition similar to that of cerebrospinal fluid. Additional obstacles are blood–perilymph and blood–strial barriers, a network of vascular endothelial cells equipped with tight junctions that greatly limits access to the inner ear following systemic administration [112]. One recent study showed that inner ear cells were transduced following the intravenous injection of the AAV2/9 viral vector only if high doses were administered via the superficial temporal vein at a neonatal stage [113]. Local routes of delivery within the inner ear, to the perilymphatic or endolymphatic space, appear to be necessary for inner ear gene therapy. Surgical delivery to the right place, without inflicting significant damage on the inner ear, is, therefore, a major challenge.

### 4.1. Delivery to the Endolymphatic Space: Cochleostomy, Endolymphatic Sac, and Utricle Administration

Delivery into the endolymphatic space, and through a cochleostomy into the scala media, has been tested in preclinical studies [30,114]. Nevertheless, this route is technically difficult and it can lead to a permanent increase in hearing thresholds in adult mice. [97], especially at high frequencies [114]. Another way to access the endolymphatic space is an injection into the endolymphatic sac, a closed sac located on the intracranial side of the petrous bone (Figure 1) and connected to the endolymphatic space via the vestibular aqueduct [115]. Few studies have investigated this route of administration [116,117,118], of which one carried out in a guinea pig showed that the injection of a recombinant AdV into the endolymphatic sac led to significant levels of transduction in the cochlear and vestibular sensory organs [116]. It may be possible to deliver therapeutic agents via the endolymphatic sac in humans, as this surgical approach is already well-established for the treatment of Ménière’s disease [119]. However, precise evaluations of the volume of therapeutic agent to be delivered, and of the pressure and administration rate, would be required to prevent endolymphatic hydrops, which could result in permanent damage to the cochlea or vestibule.

Lee et al. recently described injection into the utricle, to target the endolymphatic space, in neonatal mice [86]. Injections via this route, which is easily accessible in neonatal mice, led to a high rate of transduction in IHCs (almost 100%) and other inner ear cell types, with no damage to auditory or vestibular functions. However, the authors of this study stressed that it was difficult to ensure that the gene therapy agent was injected into the endolymph and not into the perilymphatic space. Furthermore, the endolymphatic utricular space is less easily accessible in humans, as it is almost completely covered by the facial nerve. (Figure 1).

### 4.2. Delivery to the Perilymphatic Space: Round Window, Posterior Semi-Circular Canal, and Oval Window Administration

Delivery to the perilymphatic space is the most widely used technique in preclinical studies, resulting in viral transduction rates and profiles similar to those achieved following delivery to the scala media [97]. The perilymphatic fluid can be assessed directly via two injection sites.

Injection through the RWM, which is readily accessible in both newborn and adult rodents after perforation of the tympanic bulla, results in the robust transduction of both vestibular and cochlear sensory hair cells [7,13,33,97,98]. Surgical access is more difficult in NHPs, requiring either mastoidectomy with posterior tympanotomy, a technique routinely practiced by ENT specialists during cochlear implantation [87], or trans-canal hypotympanotomy. The injection of a phosphate-buffered saline vehicle through the RWM of NHPs does not alter hearing or vestibular function [120], whereas modifications to these functions have been reported after the injection of an AAV recombinant vector via this route [33]. These findings probably reflect perilymph leakage after RWM puncture and changes in intracochlear pressure during the injection.

As a means of limiting the risk of leakage and inner ear defects, attempts have been made to administer the therapeutic gene, not by injection, but by diffusion through an intact RWM [121], potentially with the assistance of partial enzymatic digestion [98]. Although this approach results in only limited transduction of IHCs [98], and failed to restore hearing and balance in a mouse model of Usher syndrome type 1C (USH1C) [11]. Zhang et al. recently showed that ultrasound-microbubble cavitation facilitates viral transduction across an intact RWM [122]. Efforts are now being made to improve this technique, to facilitate the use of ultrasound-microbubble cavitation as an approach like any other, but with greater efficacy and fewer adverse effects, thereby optimizing the outcome of gene therapy. Furthermore, this technique would be easy to use in humans, with the possibility of administration via the external auditory canal under general or local anesthesia, for children and adults, respectively.

Another route of administration commonly used in preclinical studies is canalostomy, mostly through the posterior semicircular canal, in neonatal and adult mice [33,93,95,102]. This approach involves opening the bony canal and injecting the gene therapy agent into the perilymphatic space without damaging the membranous labyrinth. The semicircular canal is very small, rendering accurate injection difficult, and making it impossible to determine, with certainty, which compartment—perilymphatic or endolymphatic—actually received the therapeutic product. It would also be difficult to transpose the exact method to humans, due to the anatomical position of the posterior semicircular canal. Injection through the lateral semi-circular canal might be feasible in the human inner ear, provided this method had already been demonstrated to be safe in large-animal models.

By combining two techniques (RWM injection associated with canal fenestration to provide an exit hole), Yoshimura et al. greatly improved viral gene delivery to the mouse inner ear and the rates of transduction of both cochlear and vestibular hair cells, without affecting auditory function. This improvement resulted from the use of canal fenestration to create an exit path, facilitating a longitudinal flow of the injected viral preparation throughout the inner ear [101]. Interestingly, a similar delivery approach was also found to enhance virus-mediated gene delivery and inner ear hair cell transduction rates relative to RWM injection alone in NHPs. Longitudinal flow can easily be reproduced in humans, by injection through the RWM with the creation of an oval window opening, thereby significantly increasing transduction rates for inner ear hair cells, as already demonstrated in NHPs [89]. Nevertheless, it remains unclear whether this double opening of the cochlea and the resulting longitudinal flow pattern preserve vestibular and auditory function in NHPs [89].

Finally, another route of access to the perilymphatic compartment routinely used in otologic surgery is the opening of the oval window by platinotomy. This route is not used in preclinical studies in rodents because the oval window is covered by the stapedial artery in these animals, but this artery is not present in primates [123]. Only one phase I/II clinical trial for hearing loss to date has used this route of administration to deliver the viral vector, and the results of this study have yet to be reported (NCT02132130).

The perilymphatic space is likely to be the main target for clinical applications, but fluid may be exchanged between the cerebrospinal fluid and the perilymph across the cochlear aqueduct, in which case the therapeutic agent may diffuse outside the inner ear [115]. Improvements in our understanding and a clear identification of the routes of diffusion (in and out of the inner ear) of the therapeutic agent will undoubtedly help to optimize the targeting of particular cells and compartments.

### 4.3. Volume Injected and Its Flow Rate

The volume of the inner ear fluids is correlated with body mass index: the perilymph has a volume of about 0.62 µL in mice, 8.66 µL in guinea pigs [124], 26.7 µL in *Macaca nigra* [125], and 51 µL in humans [124].

In small mammals, relatively large volumes (1 to 2 µL), exceeding that of the perilymphatic compartment, are routinely locally injected in preclinical studies without generating hearing or vestibular impairment, probably because the overflow towards the CSF via the cochlear aqueduct remains open [126] preventing changes in intracochlear pressure during the injection. In macaques, there is no evidence of damage to the inner ear after administration of up to 30 µL, corresponding to more than 100% of the perilymph volume [120], suggesting clearance through the cochlear aqueduct or a leak around the injection point. NHPs, particularly subspecies from the Old-World group of monkeys, such as Macaca (Macaca fuscata, nigra, rhesus, or fascicularis), have a number of features in common with humans, including inner ear architecture and function [127], potentially allowing future extrapolations in terms of the pharmacokinetics and safety of inner ear injection [128].

## 5. Unresolved Issues

### 5.1. The Temporal Window for Therapeutic Intervention

Mouse models for human deafness have been widely and successfully used to investigate various aspects of auditory gene therapy, including delivery routes, vector specificity and efficiency, and spatiotemporally controlled gene expression. A number of preclinical investigations have been performed, which have established proof of concept for the feasibility and efficacy of gene therapy for the treatment of deafness (Table 1). However, there is still an absence of clinical trials with encouraging outcomes because the murine models for human deafness used are not ideal for predicting outcomes in patients. Indeed, mice are born deaf and do not begin to hear until the 12th postnatal day (P12); the cochlea is, thus, immature and continues to develop after birth [129]. By contrast, humans begin to hear after about four months in utero [130]. Most of the translational gene therapy studies performed in mouse models to date involved interventions performed from P1 to P9 (Table 1). The corresponding therapeutic window in humans would fall between 18 and 25 weeks of gestation, a period during which the risks of intervention are much higher, with potential safety issues relating to fetal intervention, delivery of the transgene, infection, premature delivery, and even the potential loss of the fetus. These multiple risks are the principal reason for the lack of planned trials in humans in the near future [131]. However, it should be borne in mind that patients with the most common causes of deafness potentially treatable by gene therapy are usually diagnosed during the neonatal period. The critical question is whether gene therapy shortly after birth can be effective in these patients. For any clinical application of gene therapy for deafness in humans to be considered in the near future, it will, therefore, be essential to identify the optimal therapeutic time window during which gene therapy can reverse existing deafness or prevent the progression of hearing impairment regardless of the stage already reached. A major step forward for gene therapy for deafness was taken with the study by Akil et al., 2019 [13], which addressed two of the most serious issues faced in inner ear gene therapy: the limited DNA packaging capacity of adeno-associated virus (AAV) vectors (about 5 kb, below the size of most known deafness genes), and the lack of evidence for gene therapy being able to reverse an existing deafness phenotype (cure as opposed to prevention). The authors focused on DFNB9 (MIM601071), which accounts for 2–8% of cases of prelingual hearing impairment [108,109] and is caused by biallelic mutations of the *OTOF* gene, encoding otoferlin [110,132]. Akil et al. adopted a dual AAV-vector strategy to transfer the *Otof* cDNA (~6 kb) with two different recombinant vectors, one containing the 5′ and the other the 3′ portion of the otoferlin cDNA. The authors demonstrated that local gene therapy in mutant mice not only prevents deafness when administered to immature hearing organs, but also durably restores hearing when administered at a mature stage, after hearing onset, which occurs several days after birth in mice. This finding, considered a major breakthrough, raises hopes for the possibility of future gene therapy trials in DFNB9 patients. In fact, several pharmaceutical companies are currently developing gene therapy products to treat DFNB9 patients.

### 5.2. Does the Inner Ear Have Immune Privilege?

The immune system is known to be an important determinant of gene therapy outcomes. Improvements in the design of the viral vectors most widely used in current gene therapy studies, including AdV, LV, and AAV vectors, have clearly decreased the risk of insertional mutagenesis. Nevertheless, the possibility of deleterious effects of in vivo gene therapy on the target cells or organs, including inflammation and/or immunotoxicity, remains a significant issue requiring careful consideration. There is currently insufficient evidence to support the assertion that the immune privilege of the inner ear is complete.

The eye, which is at the forefront of the field of gene therapy, has long been considered immune-privileged, but recent data have shown that viral gene transfer to the eye can trigger an adaptive immune response in both NHPs and patients [133,134,135]. A similar immunoreaction to the therapeutic vector cannot be excluded in the inner ear. The auditory hair cells are postmitotic at birth and are not subsequently regenerated. Comprehensive investigations are therefore required, to assess the risks associated with a possible immune response to inner ear gene therapy, such as damage to the treated tissue, for example, to maximize the safety and efficacy of the approach. These investigations will include evaluations of humoral and cellular immune responses, and studies of possible histological changes reflecting local immune responses in tissues of interest. Studies of these immune responses to different routes of cochlear administration will need to be carried out in both NHP and murine models.

Another issue concerns the high prevalence of neutralizing anti-AAV antibodies in humans. Seropositivity rates range from about 72% for AAV2 and 67% for AAV1 to about 47% for AAV9, 46% for AAV6, 40% for AAV5, and 30% for AAV8 [136,137]. This is a potentially serious problem, as these antibodies may completely prevent the transduction of a target tissue, rendering the treatment ineffective [138,139,140]. It will be essential to develop strategies for circumventing immune responses or preventing adaptive responses to capsid- or transgene-derived antigens through transient immunosuppression or immunomodulation, to ensure the long-term expression of the therapeutic gene.

## 6. Conclusions

Gene therapy has proved effective for preventing or treating several genetic causes of hearing impairment and/or vestibular defects in mouse models and may have potential as a curative treatment in humans. Various strategies could be adopted for the design of personalized gene therapy for patients, depending mainly on the time course of the pathophysiology of hearing loss and the presence or absence of an associated balance disorder. However, several factors must be considered before any application in humans. First, an evaluation of the therapy in mouse models, during a time window transposable to humans, is essential, because the chronological maturation of the inner ear differs between mice and humans. Second, an optimization of the viral vector, including capsid, promoter, and transgene engineering, will be required to improve transduction rates for the target inner ear cells in humans. This will involve an evaluation of inner ear cell tropism in large-animal models (especially NHPs), and in human inner ear cells. The safety of surgery, the gene delivery vector, and the therapeutic transgene will also need to be evaluated. Approaches based on injection through the RW and OW are commonly practiced by otological surgeons, but their safety must be evaluated for AAV vector-mediated cochlear gene therapy, to ensure the preservation of inner ear structure and function. Establishment of the appropriate volume of therapeutic agent and its rate of delivery is also a concern, given the small volume of the fluids present in the inner ear. Underdosage may limit AAV transduction efficiency and overdosage may result in avoidable toxicity. It is therefore essential to determine optimal dose windows for AAVs. Biodistribution studies will make it possible to evaluate the potential risk of adverse effects in cases of off-target transgene expression. Finally, the possible occurrence of an immune response to gene therapy must be considered before inner ear treatments are applied in humans. As for injections into the brain and eye, we expect the immune response to be weaker than that following systemic delivery, but preventive treatments may nevertheless be required when gene therapy agents are administered to the inner ear. The use of gene therapy to treat genetic causes of hearing impairment may become possible in the near future, paving the way for cures for several genetic causes of hearing loss.

## Figures and Tables

**Figure 1 jcm-12-01046-f001:**
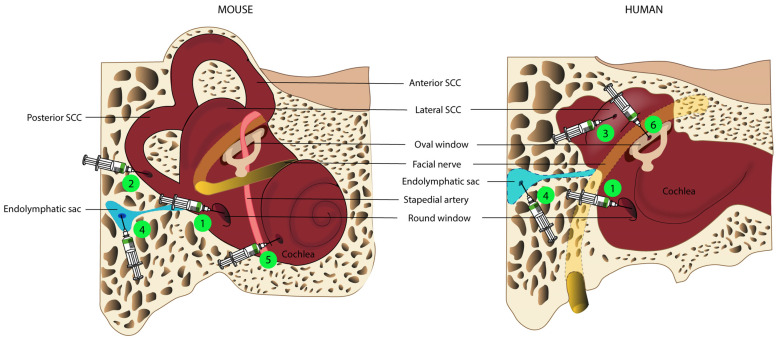
Route of administration for inner ear gene therapy. Routes of administration evaluated in mouse models (**left panel**), and potential delivery routes in humans (surgical view through the middle ear cavities, **right panel**). The most widely used delivery route is injection through the round window membrane (1), which is possible in both mice and humans, followed by the posterior semicircular canal (SCC, 2) in mice, which is potentially equivalent to a lateral SCC injection (3) in humans. Other techniques that have been evaluated in mice include injection into the endolymphatic sac (4) and cochleostomy (5). Injection through the oval window (6) is possible in humans, but not in mice, due to the persistence of the stapedial artery in rodents.

**Table 1 jcm-12-01046-t001:** Preclinical studies of gene therapy in mouse models of genetic hearing impairment.

Gene (Deafness)	Mouse Model	Stage	Approach	Vector	Strategy	Results	References
*VGLUT3* (DFNA25)	*Vglut3^-/-^*	Mature	RW	AAV2/1	Replacement	Improvement in hearing to near-normal ABR thresholds	Akil et al., 2012 [8]
		Neonatal	RW/Co	AAV2/1	Replacement		
*GJB6* (DFNB1)	*Gjb6^-/-^*	In utero	Otocyst	-	Replacement	Improvement of hearing (thresholds: 50 dB)	Miwa et al., 2013 [16]
	*Gjb6^-/-^*	Neonatal	PSCC	BAAV	Replacement	Protein production without hearing improvement	Crispino et al., 2017 [17]
*GJB2* (DFNB1)	Foxg1-cCx26KO	Neonatal	Co	AAV2/1	Replacement	Protein production without hearing improvement	Yu et al., 2014 [18]
*MSRB3* (DFNB74)	*MsrB3^-/-^*	In utero	Otocyst	AAV2/1	Replacement	Improvement in hearing to near-normal ABR thresholds	M.-A. Kim et al., 2015 [12]
*TMC1* (DFNB7/11)	*Tmc1^Δ/Δ^*	Neonatal	RW	AAV2/1	Replacement	Partial improvement of hearing (thresholds: 90 dB)	Askew et al., 2015 [19]
	*Tmc1^Δ/Δ^*			AAV2/Anc80L65	Replacement	Partial improvement of hearing (thresholds: 60 dB) Improvement of auditory cortex responses	Nist-Lund et al., 2019 [20]
	*Tmc1^Y182C/Y182C^*	Neonatal	NR	AAV2/Anc80L65	Base editing	Partial and transient improvement of hearing (thresholds: 90 dB)	Yeh et al., 2020 [21]
*TMC1* (DFNA36)	*Tmc1^Bth/+^*	Mature	RW + PSCC fenestration	AAV2/9	Regulation (miRNA)	Prevention of the progression of deafness	Yoshimura et al., 2019 [22]
	*Tmc1^Bth/+^*	Neonatal	Co	Liposome	Gene editing (CRISPR-Cas9)	Prevention of the progression of deafness	Gao et al., 2017 [23]
	*Tmc1^Bth/+^*	Neonatal	Intracochlear	AAV2/Anc80L65	Gene editing (CRISPR-Cas9)	Prevention of the progression of deafness up to one year after treatment	György et al., 2019 [9]
*PJVK* (DFNB59)	*Pjvk^-/-^*	Neonatal	RW	AAV2/8	Replacement	Improvement in hearing to near-normal thresholds	Delmaghani et al., 2015 [10]
*LHFPL5* (DFNB67)	*Lhflp5^-/-^*	Neonatal	RW	Exo-AAV2/1	Replacement	Partial improvement of hearing (ABR thresholds: 80 dB) Partial improvement of vestibular function	György et al., 2017 [24]
*OTOF* (DFNB9)	*Otof^-/-^*	P6-P7	RW	AAV2/6	Replacement	Partial improvement of hearing (thresholds: 70 to 90 dB)	Al Moyed et al., 2019 [25]
	*Otof^-/-^*	P10-P30	RW	AAV2quadY-F	Replacement	Improvement in hearing to near-normal thresholds	Akil et al., 2019 [13]
*KCNQ1* (Jervell Lange-Nielsen)	*Kcnq1^-/-^*	Neonatal	RW	AAV2/1	Replacement	Prevention of cochlear morphological abnormalities Improvement in hearing to near-normal thresholds	Chang et al., 2015 [26]
*USH1C* (Usher type 1C)	*Ush1c c.216G > A*	Neonatal	RW	Anc80L65	Replacement	Complete restoration of balance Partial improvement of hearing (thresholds: 50 dB)	Pan et al., 2017 [27]
		Neonatal	IP	-	Regulation (ASO)	Partial improvement of hearing (thresholds: 50 dB)	Lentz et al., 2013 [28]
						Complete restoration of balance	Vijayakumar et al., 2017 [29]
		Mature				Partial restoration of balance	
		Neonatal	RW/ITI	-	Regulation (ASO)	Complete restoration of balance Improvement in hearing to near-normal thresholds	Lentz et al., 2020 [11]
		Mature	ITI			Partial improvement in hearing Significant improvement in balance function	
*USH1G* (Usher type 1G)	*Ush1g^-/-^ mice*	Neonatal	RW	AAV2/8	Replacement	Complete restoration of balance Partial improvement of hearing (thresholds: 50 dB)	Emptoz et al., 2017 [7]
*WHRN* (Usher type 2D)	*Whrn^wi/wi^*	Neonatal	RW	AAV2/8	Replacement	Restoration of stereocilium structure and prevention of cell degeneration without improvement of hearing	Chien et al., 2015 [30]
		Neonatal	PSCC	AAV2/8	Replacement	Improvement of balance and hearing (thresholds: 90 dB)	Isgrig et al., 2017 [31]
*CLRN* (Usher type 3)	cClrn1KO	Neonatal	RW	AAV2/8	Replacement	Preservation of synaptic morphology with a slight improvement in hearing	Dulon et al., 2018 [32]
	*Clrn^-/-^*	Neonatal	PSCC	AAV9.PHP.B	Replacement	Partial improvement of hearing (thresholds: 40 dB for low frequencies)	György et al., 2019 [33]
*STRC* (DFNB16)	*Strc^-/-^*	Neonatal	Utricle	AAV9.PHP.B	Replacement	Partial improvement of hearing (thresholds: 40 db) with restoration of DPOAE	Shubina-Oleinik et al., 2021 [34]
*PCDH15* *(DFNB23)*	*Pcdh15^av3j^*	Neonatal	Co	AAV2/9	Gene editing (CRISPR-Cas9)	Almost complete restoration of balance Partial improvement of hearing (thresholds: 90 dB)	Liu et al., 2022 [35]

ASO = antisense oligonucleotides; Co = cochleostomy; RW = round window; PSCC = posterior semicircular canal; IP = intraperitoneal; ITI = intratympanic injection; NR = not reported.

## Data Availability

Not applicable.

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
