# Peer review of "Towards the Clinical Application of Gene Therapy for Genetic Inner Ear Diseases"

_jcm, 2023, doi:10.3390/jcm12031046_

Round 1

Reviewer 1 Report

Summarize the main findings of the study:

In the present manuscript entitled “Towards the clinical application of gene therapy for genetic inner ear diseases”, the authors aim to provide a comprehensive overview of current progress in gene therapy for inherited hearing loss. The authors summarized different gene therapy approaches, viral vectors, and delivery routes, considering their prospects and challenges to be overcome. Overall, this review is of interest and generally comprehensive, however, there are still some issues that need to be addressed, especially the lack of some recent and important research findings and clinical advances regarding gene therapy for genetic deafness.

Major suggestions:

1.     The manuscript is comprehensive but is not following up on cutting-edge research very well. Table 1 and the "Gene therapy strategies" section, it does not summarize the advances in gene therapy in the last two years regarding hereditary hearing disorders, especially gene editing for deafness treatment. Precise corrective therapeutic strategies based on CRISPR-Cas technology, including Cas9 nuclease-mediated HDR, base editors, and prime editors, are considered to be a more desired strategy as they can precisely correct mutation sites, while the authors did not mention some of these researches or did not highlight these contents.

2.     The factors affecting the transduction rate and the cell types transduced in the inner ear of AAV are summarized by the authors in a slightly general way and need to be compared and discussed in depth, otherwise, it is difficult for the reader to draw valid conclusions from them. Also, recent advances in delivery vectors in the inner ear need to be added, for example, the AAV-PHP.eB, which has been widely used in recent years, and the recently developed AAV-ie, which has high transfection efficiency in both hair cells and supporting cells, are not included.

3.    In this review, the authors emphasize the need to summarize this aspect of the progress of clinical studies of inner ear gene therapy. Usually, in preclinical studies of gene therapy, readers are particularly interested in the efficacy and safety of delivery vectors, delivery routes, etc. in non-human primate models; however, this manuscript lacks the findings of inner ear gene therapy studies in non-human primate models.

4.     The authors mentioned translational research studies or clinical applications in the manuscript many times. However, the manuscript did not review recent advances in clinical trials (e.g. DFNB9 gene therapy) in gene therapy for genetic hearing loss.

Minor tips:

1.     It is a bit confusing that in the list of authors listed, the last word is "and".

2.     The title “genetic inner ear diseases” might not be equal to the manuscript focused on “genetic deafness”.

3.     Line 15: There might have been recognized 150 deafness genes.

4.     Line 62: Gene replacement is another name for gene augmentation, so it seems not accurate by side-by-side listing; it might be better to put one of them in parentheses.

5.     There are many misquotes in the manuscript, especially in Table 1. Please check the cited references again.

6.     Whrnwi/wi and Clrn-/- line in Table 1: the approach should be PSCC, not PSSC or PSCSC.

7.     In the “Routes for inner ear gene delivery” section, it would be better to overview what routes are included in the two subtitles—delivery to the endolymphatic space and perilymphatic space in the first paragraph of this section or list behind the subtitle for readers to read.

8.     Line 50: “treatment efficacy, administration techniques, and safety” not “treatment efficacy, administration techniques and safety”.

9.     Line 162: “as vectors for” not “as vector for”.

10.   Line 187: “Only two lentivirus-based vectors have been approved by the United States Food and Drug Administration, whereas there are 209 clinical trials in humans with AAVs targeting different disorders”. It is obvious that the numbers described are wrong.

11.   Figure 1 legend: Routes for mouse model are in the left panel and human in the right panel.

12.   Some papers classified canalostomy as a delivery route into the endolymph. E.g., Milestones toward cochlear gene therapy for patients with hereditary hearing loss. 2021. How to define it?

13.   Can you provide references to support line 311-319?

Reviewer 2 Report

The topic is very important and timely considered.

Comments:

It is known that mutations of STRC gene in humans causes the hearing threshold elevation from 40 to 55 dB which corresponds to loss of outer hair cells capability to amplify sounds. It is not clear if it is necessary to treat the mild and moderate hearing loss with prenatal gene therapy.

It is necessary to check the correspondence of citations numbers with the reference list. For example:

P.2 line 48  - reference in bracket 7-3? Is it correct?

P.5 line 107 - correct reference is 28 (Maeda et al). 

The reference Chien WW 2015 in table 1 corresponds to reference 85 but not 28,

Yoshimura et al. 2019[20], but in references it is 23,

Yoshimura et al. 2018[93], but in references it is 100.

P.6 line 147-155 and further - check the reference list

Reviewer 3 Report

Review-2023-JCM-2146115-gene therapy-deafness

Towards the clinical application of gene therapy for genetic inner ear diseases

By

Lahlou G et al

This review paper is an interesting summary of the state of results from animal experience with gene therapy  strategies for dominant and recessive hearing impairment/deafness. Despite important advances to overcome existing challenges/difficulties, important remaining hindrances exist.

The paper includes tables, in which  I carefully checked the references (first Author names and numbers) and found numerous inconsistencies between those when checking with the list of  references. In the cases in the text where the name of first author and number (in the reference list) were given   (in most instances only a number was written in the text and referring to the list of references), I also noticed numerous inconsistencies. Something must have gone completely wrong and I find it impossible to further review the paper until these issues have been corrected.

Author Response

We have checked and corrected the correspondence of citation numbers throughout the manuscript in this revised version.

Round 2

Reviewer 2 Report

I recommend paper for publication

Reviewer 3 Report

the authors have corrected the many inconsistencies and missing references and I have no further comments